# Ranking of Illegal Buildings Close to Rivers: A Proposal, Its Implementation and Preliminary Validation

**Paolino Di Felice** 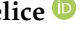

Department of Industrial and Information Engineering and Economics, University of L'Aquila, 67100 L'Aquila, Italy; paolino.difelice@univaq.it; Tel.: +39-320-423-2040

**Abstract:** Illegal buildings (IBs) are a dramatic problem in developing countries due to the population explosion, but, at the same time, they represent an unsolved issue in several states usually called advanced (as, for instance, Italy). To protect the environment, and hence, people, land authorities must respond to the challenge of IBs by demolishing them. However, in countries where the phenomenon is extended, it is indispensable to provide those figures with an IT tool that returns to them an order of demolition. Through remote sensing methods, suspicious buildings can be identified with a good approximation, but they are all ex aequo. The research summarized in this paper formalizes a two-steps method to deal with a specific category of IBs, namely, those that are close to rivers. These buildings are of special interest to land authorities because people living or simply working inside them are exposed to the flood hazard that each year claims many victims all over the world. The first step of the method computes the census of the IBs located close to rivers, while the second step computes the ranking of these buildings. The ranking may be used as the IBs demolition order. In the paper, it is also proposed the structure of a Spatial DataBase (briefly, SDB) that is suitable to store the input data necessary to solve the problem, as well as the final ranking. Spatial SQL queries against the SDB implement the proposed two-steps method. A real case study was carried out to make a preliminary validation of the method.

**Keywords:** illegal building; flood; Spatial DataBase; SQL; metric; ranking

## 1. Introduction

This paper is about illegal buildings (IBs) located close to rivers. Those buildings are of special interest to land authorities because people living or simply working inside them are exposed to the flood hazard that each year claims many victims in many countries all over the world. "Informal settlements" is an alternative denomination of IBs. UN-Habitat calls informal settlements as follows: "(i) Residential areas where a group of housing units has been constructed on land to which the occupants have no legal claim or which they occupy illegally; (ii) unplanned settlements and areas where housing is not in compliance with current planning and building regulations".

The presence of IBs on a given territory has a long list of negative implications briefly listed below:

- IBs impact the property market, because they discourage investments in real estate development;
- IBs impact on the government's ability to manage and plan land use;
- IBs cause reduction of the revenue of the local government, because owners of those dwellings do not pay property taxes;
- IBs determine the degradation of the landscape, the primary source of revenue for countries like Italy thanks to the tourism;
- IBs contribute to expand the corruption;

- IBs are more exposed than legal constructions to natural hazards such as earthquakes (IBs have not passed any test of compliance with the rules about building stability, in other words, there is a high probability that they are structurally unsafe constructions) and floods (a problem that affects IBs built in the catchment of rivers). This issue is particularly severe because it is linked to the safety of the occupants of those buildings. In 2018, different Italian regions—Liguria and Sicily above all—suffered severe floods with damage to buildings and many victims. At the beginning of November 2018, the wave of bad weather caused 12 victims in Sicily.

This work proposes a metric ($S$), in response to two laws promulgated by the Italian Government to combat the IBs phenomenon: Law no. 42 of 2004 (*Codice dei Beni Culturali e del Paesaggio—National Code for Cultural and Landscape Heritage*) and Law no. S 580-B of 2018 (*Disposizioni in materia di criteri per l'esecuzione di procedure di demolizione di manufatti abusivi—Provisions about the criteria for the execution of the procedures for the demolition of illegal buildings*).

Article 142 of Law no. 42 fixes in 150 m the width of the so-called Strip of Respect (*SofR*) around rivers, that is, the ribbon of land where it is forbidden to build. The prohibition applies to both sides of rivers and obviously concerns new constructions. Many countries have a law that fixes the width of the *SofR*.

Law no. S 580-B requires that land authorities demolish the IBs located in their area of competence by following a demolition order of those constructions based on a large set of "objective" criteria. The hope of the legislator is that such an order might discourage owners' legal actions, whose completion usually takes many years because of the overload of the Italian Justice.

With respect to Law no. 42, the basic task to be accomplished is to take the census of the IBs, while by using metric $S$, it is possible to rank them. The ranking relates the unauthorized urbanization to the flood hazard to which people being in those constructions are exposed. With respect to Law no. S 580-B, metric $S$ is a candidate to become one of the criteria on which to base the ranking calculation of the IBs to be broken down.

Another contribution of this work concerns the structure of a *Spatial DataBase* (briefly, *SDB*), where the input datasets of the problem as well as the values of $S$ can be stored. The latter values are implemented as SQL spatial queries. The implementation is general, therefore, it can be migrated to any study area, provided that the SDB maintains the same structure proposed in this article. Moreover, the proposal can be integrated into the Land Information System today adopted by most land planners (e.g., [1,2]).

The paper is structured as follows. Section 2 (a) focuses on the relevance of the problem addressed by the research; (b) introduces definitions and notations that will be used throughout the paper; (c) formalizes the two-steps method proposed to rank the IBs located close to rivers; (d) introduces the reference software architecture adopted to implement the method and lists the tables of the underlying SDB. Section 3 reports about a case study concerning an Italian region. The case study was used to make a preliminary validation of the method. Section 4 points out the weaknesses of our method.

The SQL implementation of the proposal is given in an external *Annex* (see Supplementary Materials).

This paper extends a previous one [3] as follows: First of all, by expanding the section about the related work. Then, by formalizing the method and implementing it. Finally, by adding a real case study.

## 2. Materials and Methods

### 2.1. Relevance of the Problem

Kundzewicz et al. [4] pointed out that in the near future in Europe the risk of flooding will increase. To limit the damages caused by this hazard, it will be fundamental to (a) make urban planning that takes into account this increased hazard—a suggestion emerged also from other studies (e.g., [5]); (b) increase the severity of intervention against the IBs.

Many studies have investigated the impact of urbanization on flood events (e.g., [5–15]). According to the findings in the study by Agbola et al. [6], besides prolonged rainfalls and river overflows, there are *anthropogenic factors* of flooding. Uncontrolled building construction is a relevant factor because such constructions obstruct the free flow of water (and, as a consequence, they are at high risk of being flooded). Reporting about the August 2011 flood in Ibadan (Nigeria), Agbola et al. [6] say that in the city there were 600 buildings close to the riverbanks.

The remainder of this section focuses on the phenomenon of IBs. In recent decades, municipalities and governments in all parts of the world have been struggling against IBs. The problem of IBs is becoming dramatic in developing countries due to the population explosion (a case study about informal settlements in Bandung, Indonesia, is reported by Jones, [16]. References [17–19] are further papers on the same tipic). In [19], Qian reported that in 2013, in Shenzhen (China), the IBs were 273,000, covering about 43% of the total construction area. Unfortunately, IBs are an unsolved issue also in the so-called *advanced* states. This is the case of Italy (e.g., [20,21]).

In 2007, LegAmbiente, an Italian nongovernmental organization, published the results of a nationwide study [22]. In the report, it is mentioned that 402,676 IBs were built in Italy from 1994 to 2003. In 2018, LegAmbiente published the results of another study, [23]. The data refer to the registered violations (i.e., the infringements for which a demolition sentence was issued) in *all* the Italian regions. The report covers the period from 2005–(June) 2018 (in 2004, in Italy there was the last building amnesty). 57,432 is the total number of registered infringements. Table 1 (taken from [21]) gives the index of IBs in Italy (i.e., the ratio between the number of unauthorized buildings to the number of building permits issued by municipalities). The relevance of detecting IBs in Italy was stressed also by Cialdea and Quercio [24] with a case study concerning illegal settlements in the city of Campobasso (the capital of the Molise region, South of Italy) and its hinterland.

**Table 1.** The index of illegal buildings (IBs) in Italy.

|                  | Number of Regions | IB Rate (%) |
| ---------------- | ----------------- | ----------- |
| North            | 7                 | 5.3         |
| Centre           | 7                 | 11.6        |
| South & islands  | 6                 | 35.9        |

So far, several methods have been proposed for automatic building detection from high-resolution remote sensing images ([25–29]); few of them are specifically focused on IBs detection (e.g., [30–34]). Soon, most of them will be available on the marketplace as a plugin of GIS software. This is the easiest and fastest way to produce cyclically new datasets about the study area of interest to keep the underlying SDB updated.

Today, many urban planning departments all over the world use a Land Information System software to deal with IBs; for example Yang et al. [33] describe the architecture of one of them. A relevant part of those systems is the SDB storing city maps in the vector data format. Our proposal can be seen as an extra functionality of these systems, since it operates on data stored into the SDB.

In order to protect the environment, and hence people, land authorities must respond to the challenge of IBs by demolishing them. However, where the phenomenon of constructions against the law is extended, it is essential to help land managers with an IT tool that guides them in prioritizing demolitions. Using remote sensing methods, the suspicious buildings can be identified with a good approximation (even if rarely with 100% accuracy), but all them remain ex aequo. Returning to the land authorities, the ranking of the IBs overcomes this shortcoming.

In this paper, we give a contribution to the solution of the problem of ranking the IBs to be demolished located close to rivers, in connection to two Italian laws. The laws we refer to are Law no. 42 of 2004 and Law no. S 580-B of 2018. Our proposal is general, so it can be adapted to the laws of other countries.

As already mentioned, Article 142 of Law no. 42 fixes in 150 m the width of the *SofR* around rivers. Law no. 42 is relevant because it implements the point of view of many scholars about the fact that to limit the losses due to the more frequent and violent floods, the attention of the academia and, hence, the attention of land authorities has to be shifted away from *protection* towards more emphasis on *mitigation*. The findings of scholars such as, for instance, Burby [35] and Holway and Burby [36] are on the same line. In fact, they have pointed out that to limit the negative effects of extreme flood events, it is necessary, first of all, to restore river spaces to their original state and, then, to add an extra *horizontal* space. The importance of procuring horizontal river spaces rather than reinforcing embankments has been reiterated recently by Ha and Jung [37] as the more effective solution in the long-term to prevent damage to buildings and, therefore, to the people living or working inside them. In their paper, they call *conservation easement* the strip of land of predetermined size along a river that remains of public ownership. The notion of conservation easement is equivalent to the concept of *rolling easement* proposed by McLaughlin [38] years before as a conceptual tool to protect sea coastlines. The conservation/rolling easement notion is a simple and at the same time flexible mechanism whose implementation and success is highly dependent upon a clear legislative action. In the case of Italy, Law no. 42 is the expression of such intent.

The *SofR* of Law no. 42 is a "rigid" implementation of the notion of *freedom space for rivers* that constitutes the basis of the work by Biron et al. [39]. Those authors' hope is that such a notion will "be implemented in future river management legislation because it promotes a sustainable way to manage river systems, and it increases their resilience to climate and land use changes in comparison with traditional river management approaches which are based on frequent and spatially restricted interventions.", ([39], p. 1056). Biron et al. proposed a hydrogeomorphic approach to delimit the freedom space around a river. Their method requires a combination of geographical information system analysis and field observations. In the meanwhile that the countries in the world make available the freedom space map for each of the rivers that cross them, the easiest solution consists of equipping land authorities with a law like Law no. 42, adopted by the Italian government a long time ago.

Very recently, Pathak and Ahmad [40] investigated the fundamental role that the government of a country can play after a flood disaster event has occurred, to improve the recovery process. However, evidently, the governments must also play an active role in preventing or at least in mitigating the damage caused by a flood disaster. The Italian Law no. 42 also plays this role.

Law no. S 580-B lists the set of general criteria that the land authorities have to take into account in the definition of an objective order of demolition of the IBs present over their administrative unit (either a municipality, a province or a region). The criteria in Article 1 of such a law are recalled below.

*Environmental issues* They regard IBs of either (a) relevant environmental impact or built on state-owned area or (b) that violate an environmental constraint (i.e., IBs built on an area subject to either environmental, landscape, seismic, hydrogeological, archaeological, or historical-artistic constraint). Law no. 42, for example, fixes an environmental constraint.

*Dangerousness for (public and private) Safety* This criteria imposes to assess to which degree an IB may represent a danger for the safety. Aspects to be taken into account concern the deterioration of structural elements and the danger of collapse of nonstructural parts. Moreover, it is important to take under consideration if the IBs were realized without a structural design.

*Realization Phase* IBs that are still under construction must be demolished before the finished ones.

*Destination of Usage* Examples of pertinent options are as follows: IBs used by convicted subjects; IBs used for illegal economic activities; IBs used as second/holiday home; IBs used for commercial/industrial activities; IBs used as unique dwelling of a poor family without/with children.

*Data of Crime* Older IBs have to be demolished first.

Identifying a demolition order that makes multiple conflicting criteria coexist falls into the category of problems called *Multiple Criteria Decision Making* problems—MCDM (e.g., [41]) or *Multiple Attribute Decision Making* problems (e.g., [42]). Among the methods proposed for the solution of MCDM problems, the so-called *Technique for Order Preference by Similarity to Ideal Solution* (*TOPSIS*) assumes

an important role. TOPSIS was proposed by Hwang and Yoon [43]. Forte et al. [44] have addressed the problem of calculating the demolition order of the IBs located in Italy with reference to a draft version of Law no. S 580-B (in their paper, called Bill no.580-A), by using TOPSIS. In this frame, metric *S* may be used as one specific category of environmental constraints, namely, that set by the Italian Law no. 42.

### 2.2. Notations

Hereinafter, we use the following notations:

*GeoArea* is the portion of land of interest for the study (e.g., a municipality, a region, or a state). *GeoArea* is defined as the pair ⟨*description, geometry of the boundary of the GeoArea*⟩, where *description* is a string.

$\mathcal{C} = \{c_g(g = 1, 2, ..., card(\mathcal{C}))\}$, where $c_g$ is a *contour line*, which is a curve whose points have the same elevation with respect to the sea level. A generic contour line is defined as the tuple ⟨*ID, elevation, geometry*⟩, where *ID* is an identifying code.

$\mathcal{R}$ *(Rivers)*= $\{r_k(k = 1, ..., card(\mathcal{R}))|r_k$ is a *river* that crosses the *GeoArea*}. The generic river is described by the tuple ⟨*ID, name, geometry*⟩. *RiverBuffer*($r_k$) denotes a buffer of width *w* around river $r_k$; the buffer is the geometric counterpart of the legal notion of *SofR*.

$\mathcal{B} = \{b_i(i = 1, 2, ..., card(\mathcal{B}))\}$, where $b_i$ denotes a *building* in the *GeoArea*. Each building in $\mathcal{B}$ is defined as the tuple ⟨*ID, geom, status, elevation, S*⟩, with *geom* being the footprint of $b_i$, *status* is a Boolean variable denoting whether the building is illegal or not, and *elevation* is the value of the building's altitude over the sea level. *S* is a positive numeric value denoting the degree of (spatial) exposure of $b_i$ to the flood hazard.

### 2.3. The Metric S

In the following, first, it is shown *how* to detect IBs inside the *GeoArea*, then, it is introduced the metric *S* to rank the severity of the violation.

#### 2.3.1. Step 1: Census of IBs

The buildings that violate Law no. 42 are those for which it happens that there is a nonzero intersection between the building footprints and *RiverBuffer*($r_j$).

#### 2.3.2. Step 2: Ranking of IBs

Equation (1) introduces the (dimensionless) parameter *P* that measures the extent of the penetration of the footprint of $b_i$ into the *SofR* of the generic river $r_j$, whose width (*w*) is established by the law (*w* = 150 m in the case of Italy). *P* = 0 denotes the absence of violation, while any other value of *P* in the range (0,1] denotes the opposite.

$$P = \frac{Area(geom(b_i) \cap RiverBuffer(r_j))}{Area(geom(b_i))}. \tag{1}$$

Equation (2) introduces the metric *S* that attributes to each IB a decimal value (greater than zero) that measures its degree of exposure to the risk of flooding, following prolonged periods of rain. For this reason, it is correct to state that such a metric suggests the priority of demolition of the IBs, in order to prevent casualties.

$$S = max(P_k/d_k) \times \begin{cases} 1 - \Delta h_{b_i} & \text{, if } \Delta h_{b_i} < 0 \\ 1/(1 + \Delta h_{b_i}) & \text{, otherwise.} \end{cases} \tag{2}$$

In Equation (2), *d* denotes the minimum (Euclidean) distance between the boundary of building $b_i$ and the geometry that models the river. If *d* is less than 1 m, then *d* is set to 1 to avoid the *division by zero*.

Geography teaches that rivers can have tributaries. For buildings that are near the points where two waterways merge, it can happen that their geometry overlaps the $SofR$ of both waterways. Therefore, for the same building, we can have $k \in [1, 2, ..., n]$ values for the ratio between $P$ and $d$; with t$n$ being the number of intersections building–river.

$\Delta h_{b_i}$ denotes the elevation difference, in meters, above sea level, between the centroid of building $b_i$ and the geometry of the riverbed, at the point of minimum distance between those two geometries. The elevation of a point is its height above sea level; while the elevation variation of building $b_i$ (described by its centroid) with respect to the river $r_j$ (described by the point, belonging to the river geometry, most closed to $b_i$) is the value, taken with sign, of the difference given by the equation:

$$\Delta h_{b_i} = h_{b_i} - h_{r_j}. \tag{3}$$

The two alternatives of Equation (2) on the right side of the left brace take into account the territory elevation inside the $GeoArea$. Figure 1 shows four different configurations of hypothetical IBs, in pairs of two. Below, we comment the two pairs of geometric scenes, in sequence.

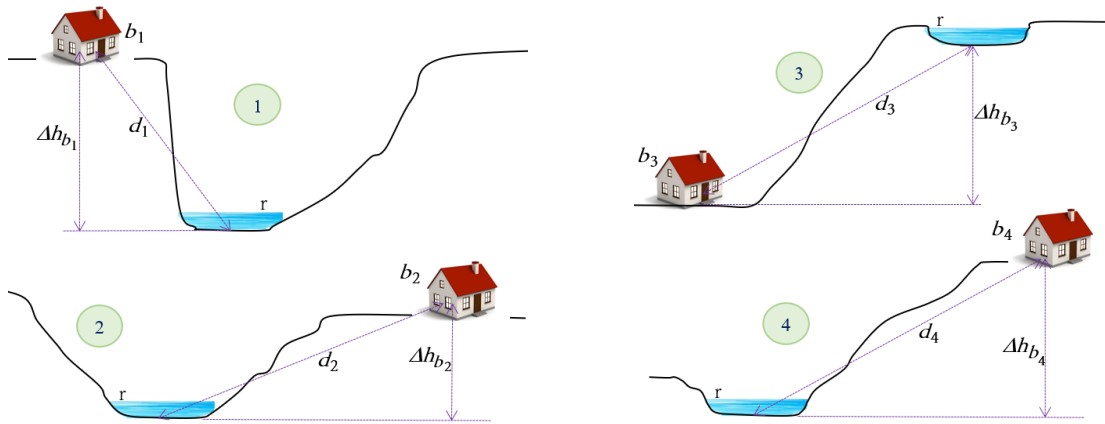

**Figure 1.** Four IBs.

Figure 1 (left)—Buildings $b_1$ and $b_2$ satisfy the following: $d_1 < d_2$, while $\Delta h_{b_2} < \Delta h_{b_1}$ ($\Delta h_{b_1}$ and $\Delta h_{b_2}$ are both positive). With respect to the flood hazard due to the river ($r$), building $b_2$ is more exposed than $b_1$ as the barrier separating it from the river is lower than the barrier separating $b_1$ from the river.

Figure 1 (right)—Buildings $b_3$ and $b_4$ satisfy the following: the values of $d_3$ and $d_4$ and the values of $\Delta h_{b_3}$ and $\Delta h_{b_4}$ are comparable ($\Delta h_{b_3}$ and $\Delta h_{b_4}$ have a discordant sign; negative and positive, in order). The values of $d_1$ and $d_4$ are comparable too, while $\Delta h_{b_3} < \Delta h_{b_1}$. With respect to the flood risk due to the river ($r$), building $b_3$ is more exposed than $b_4$ since there is no protective barrier in case of river flooding, unlike what is observed for building $b_4$.

In summary, it follows that the ranking that correctly reflects the degree of exposure to the flooding hazard of the four IBs of Figure 1 is $b_3$, $b_2$, $b_4$, and $b_1$. Equation (2) returns the expected result, if applied to the four configurations of Figure 1.

The two alternatives of Equation (2) on the right side of the left brace correct the value of the term $max(P_k/d_k)$ by taking into account the height difference between the centroid of building $b_i$ and the geometry of the riverbed, at the point of minimum distance between the two geometries. In fact, those two alternatives amplify the value of term $max(P_k/d_k)$ when the IB is located below the river, while reducing it in the opposite case. The final effect on the ranking is that the buildings below the level of the river raise towards the $top$ positions, while the buildings above the level of the river slide in the $queue$.

*2.4. Implementation of the Proposal*

Figure 2 shows the reference software architecture used to implement the method summarized in the previous subsection. The *Spatial DataBase Management System* (SDBMS) is the basic element of the architecture. It is on top of an SDB where the input data of the problem as well as the ranking about the IBs in the *GeoArea* are stored. It is well-known that the SDBMS technology offers enormous advantages to processing geographic data. PostgreSQL/PostGIS was adopted as the software technology.

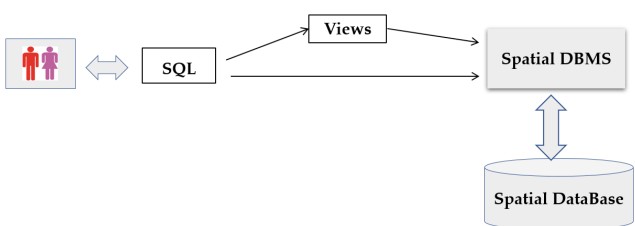

**Figure 2.** The reference software architecture.

Table 2 shows the mapping of the theoretical concepts into database entities.

**Table 2.** Entity mapping.

| Definition | Entity |
| --- | --- |
| *GeoArea* | GeoArea |
| $\mathcal{C}$ | ContourLines |
| $\mathcal{R}$ | Rivers |
| *Buildings* | Buildings |

The four tables of the SDB are listed below. The underlined attribute denotes, as usual, the *primary key* of the table it belongs to.

- GeoArea(<u>id</u>, geom);
- ContourLines(<u>id</u>, elevation, geom);
- Rivers(<u>id</u>, name, geom, river_buffer);
- Buildings(<u>id</u>, geom, status, elevation, S).

Within the setting of Figure 2, the computation of metric *S* was quite trivial, in fact, it took place as SQL spatial queries. To keep few of them easy, we made recourse to the SQL *view* abstraction (Figure 2).

The SQL scripts of the implementation are given in an external Annex (see Supplementary Materials).

## 3. A Case Study

In Italy, well-known populous cities are located close to important rivers. Florence is an example. It is crossed by the Arno river. In November 1966, Florence was overwhelmed by a violent flood. The balance: 17 victims, enormous damage to the city and its artistic heritage (Figure 3). Thousands of volumes, including precious manuscripts or rare printed works, were covered with mud in the Central National Libraries. Countless were the damages to the Uffizi museum. To prevent the recurrence of mournings caused by events such as the one just mentioned, many laws have been promulgated in Italy such as the already mentioned Law no. 42.

This section reports about a case study we have carried out to make a preliminary validation of metric *S*. To achieve an effective and complete presentation of the results by means of maps, we put the *QGIS* software beside PostgreSQL/PostGIS. QGIS guarantees full compatibility with PostGIS.

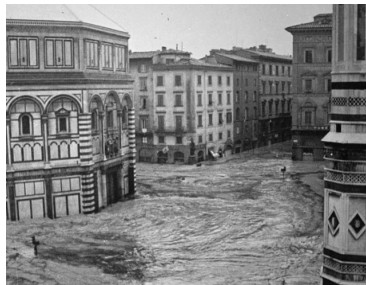 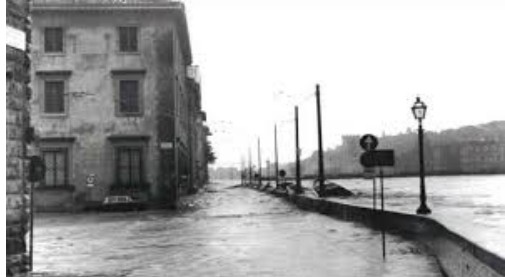

**Figure 3.** Two images about the flood of Florence (1966).

### 3.1. The Input Data

*GeoArea*. It coincides with the boundary of the Abruzzo region (Figure 4), an area of 10,800 km$^2$ and 1,330,000 inhabitants.

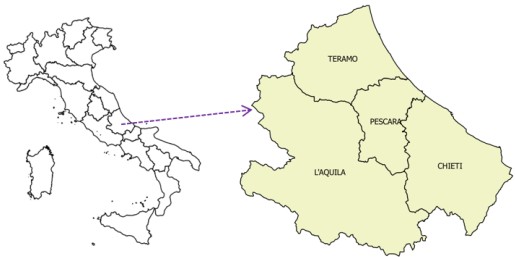

**Figure 4.** The *GeoArea*: the Abruzzo region.

$\mathcal{C}$. The contour lines of the Abruzzo territory have been generated, starting from raster data about such an Italian region. The second step consisted in constructing, through QGIS, a shapefile of the contour lines of the Abruzzo extracting them from the just mentioned raster. The elevation interval between two adjacent lines was set to 5 m. For the problem being studied, it is important to have contour lines very close to each other. The third and last step consisted in intersecting the contour lines returned by QGIS with the geometry of the boundary of the Abruzzo region (the *GeoArea*). Figure 5 (left) shows the map about the contour lines of the Abruzzo region.

$\mathcal{R}$. The shapefile about the rivers that cross the Abruzzo territory has been downloaded from the portal of ISPRA (*Istituto Superiore Protezione e Ricerca Ambientale—Institute of Protection and Environmental Research*). Figure 5 (right) shows the map about those rivers.

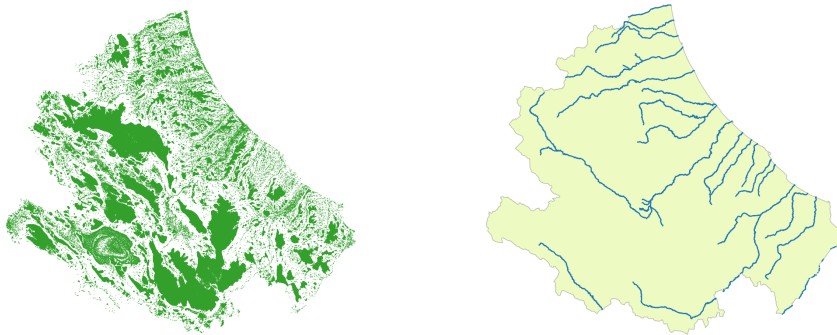

**Figure 5.** The contour lines (**left**) and the rivers (**right**) of the Abruzzo's region.

$\mathcal{B}$. The shapefile about the buildings that are inside the boundary of the Abruzzo region has been downloaded from the website http://download.geofabrik.de. The dataset consists of 90,381 records. Figure 6 shows a portion of the buildings of the city of L'Aquila, the capital of the region, and the two rivers (*Aterno* and *Raio*) in the area.

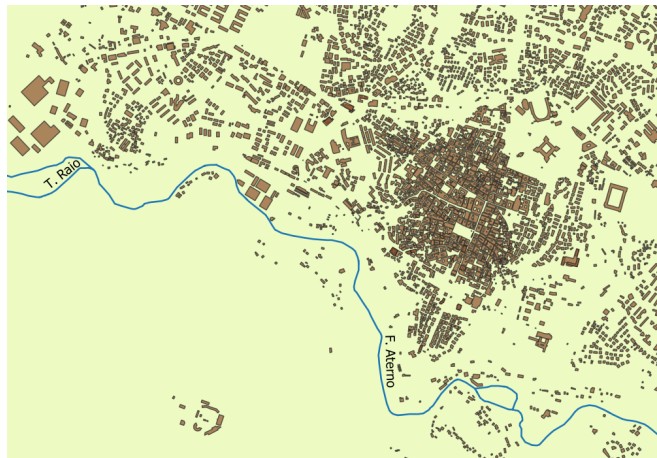

**Figure 6.** Buildings in the city of L'Aquila.

*3.2. Results*

3.2.1. IBs Census

2233 buildings out of 90,381 (i.e., the 2.47%) violate Law no. 42, so they are illegal and, hence, have to be removed. 2233 is a big number; that is why it is fundamental to provide land authorities with an order of demolition.

3.2.2. Ranking of the IBs

The computation of metric $S$ returned the ranking of the IBs. The map of Figure 7 shows few IBs. In the map, the IBs are clustered into three classes: *high*, *medium*, *low*, denoted with the colors red, orange, and yellow, respectively. The map shows, moreover, the distribution of the IBs inside those three classes, whose extreme values were arbitrary fixed.

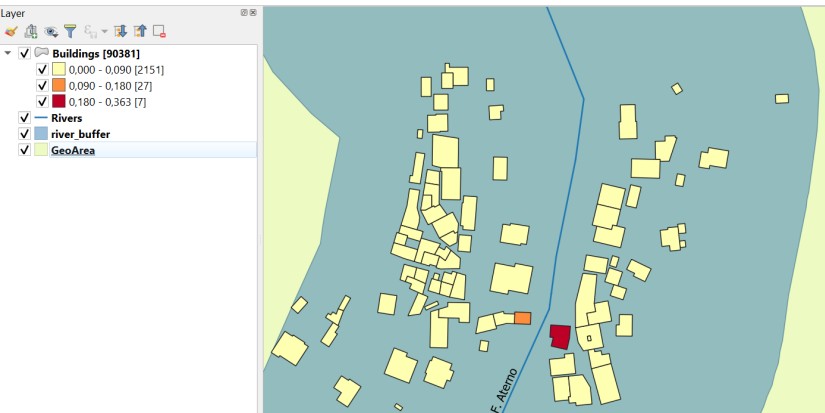

**Figure 7.** Few IBs ranked according to metric $S$.

Figure 8 (left) shows the top-7 IBs according to the value of metric $S$. The figure shows, moreover, for each IBs the WGS84 coordinates of its centroid, the value of $S$, the elevation, and the difference of elevation with respect to the closest river. The map of Figure 8 (right) shows the location of the top-7 IBs inside the Abruzzo region, as well as the corresponding $S$ value.

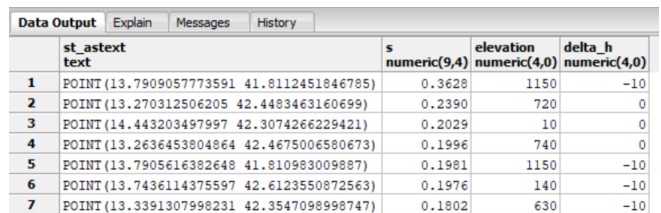

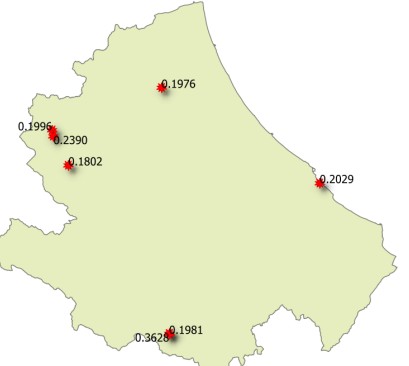

**Figure 8.** The top-7 IBs ranked according to metric *S*.

### 3.3. Validation of the Ranking

In the Machine Learning domain, the notion of "confusion matrix" was introduced to assess the performance of the classification algorithms with respect to some test data. It is a two-dimensional matrix, indexed in one dimension by the "true class" of an object and in the other by the class that the classifier assigns [45].

The simplest way to validate the ranking based on metric *S* is to reduce the problem of the classification of experimental values to the case of *binary confusion matrices*, that is, to the case in which only one class at a time is involved. Therefore, the validation problem can be formulated as follows.

*Given n values ($v_1$, $v_2$, ..., $v_n$) and a class (C), construct the binary confusion matrix that summarizes how those values are classified both in the real world and as estimated by the algorithm of which we want to "measure" the effectiveness. Evidently, the value $v_1$ may fall in C or not, the same holds for $v_2$, ..., $v_n$.*

The main diagonal of the confusion matrix has the number of elements predicted correctly. The total number of elements belonging to a real class is equal to the sum of the values on the corresponding row of the table. The total number of the elements present in the set involved in the classification operation is equal to the sum of all the totals.

Table 3 shows the structure of a generic binary confusion matrix. In the following, the terms true positives, false positives, true negatives, and false negatives are defined (they are positive integers) [45]:

- *True Positives* (*TP*). This quantity denotes the cases in which the classification algorithm has recognized *correctly* the class to which they belong to.
- *False Positives* (*FP*). This quantity denotes the cases of *wrong classification*. In practical terms, a false positive constitutes a *false alarm*.
- *True Negatives* (*TN*). This quantity denotes the cases that the algorithm has recognized *correctly* not belonging to the class.
- *False Negatives* (*FN*). This quantity denotes the cases for which the algorithm has confused the class to which an element belongs to.

Many metrics have been proposed to judge the goodness of an algorithm that reconstructs the observed reality ([46] is an authoritative source on the subject). The most common of them are listed below. The value of the metrics expresses a *marginal probability* between 0 and 1.

**Table 3.** Structure of a binary confusion matrix.

|  |  | Predicted Class | | Total |
|---|---|---|---|---|
|  |  | Class | No Class |  |
| Actual Class | Class | TP | FN | P = TP + FN |
|  | No Class | FP | TN | N = FP + TN |
|  | Total | TP + FP | FN + TN | P + N |

*Recall* is defined as the percentage of positive cases correctly recognized as such (by the adopted classification method):

$$Recall = \frac{TP}{TP + FN}. \tag{4}$$

*Precision* is defined as

$$Precision = \frac{TP}{TP + FP}. \tag{5}$$

*Accuracy* is defined as

$$Accuracy = \frac{TP + TN}{P + N} = \frac{TP + TN}{(TP + FN) + (TN + FP)}. \tag{6}$$

### 3.3.1. Manual Classification of the IBs

Classifying 2233 IBs by hand is hard. We limited the classification to the 34 IBs that fall into the classes *High* and *Medium* of Figure 7 (below, briefly called the top-34 IBs). The classification was carried out by means of the QGIS geobrowser and Google Earth, whose motto is "*Travel the world without getting up from your seat*". Each IB was classified, with respect to its level of exposure to the flood hazard, by taking into account its proximity to the closest riverbed and the contour lines of the territory.

Figure 9 shows two images taken from Google Earth. The picture on the left shows the Abruzzo region; while that on the right shows the IB in the second position in the ranking of Figure 7. This IB is very close to the Aterno river that, in the picture, is almost covered by the trees.

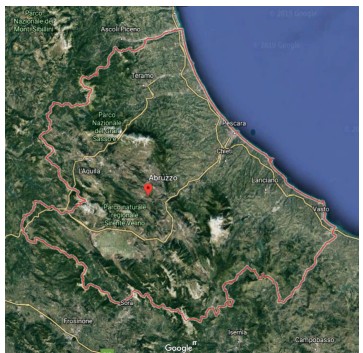 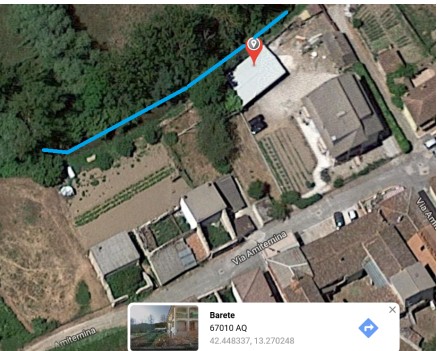

**Figure 9.** Two images taken from Google Earth. The blue line highlights the river hidden by the trees.

Drawing inspiration from a widespread practice for hazard classification studies (e.g., [47]), three hazard classes were adopted: *High*, *Medium*, and *Low*. The IBs located *very close* to the riverbed and at the same time located either *below* the river elevation or on a flat terrain were placed in the High-hazard class, while the IBs located *near by the boundary* of the river buffer and at an elevation *higher* or equal than it were placed in the Low-hazard class. The Medium-hazard class was attributed to the hybrid situations. Table 4 shows the result of the manual classification. In Table 4, the IBs are identified by their ID. The match between the ID and the geographic coordinates is shown in Table 5.

**Table 4.** Manual classification of the IBs.

| Class | Number of IBs | List of Their ID |
|---|---|---|
| High | 7 | 35990, 66089, 35986, 73424, 87942, 70015, 16044 |
| Medium | 27 | 6611, 16013, 16048, 24450, 26999, 33206, 45612, 45979, 45980, 45981, 45982, 46250, 48394, 52737, 70014, 73283, 73375, 73430, 74941, 74943, 87075, 87079, 87081, 87083, 87084, 87114, 87927 |
| Low | 0 | |

**Table 5.** ID, geographic coordinates, and score of the top-34 IBs.

| ID | The WGS 84 Coordinates of the Top-34 IBs | The Score |
|---|---|---|
| 35986 | 13.7909057773591 41.8112451846785 | 0.3628 |
| 70015 | 13.2703125062050 42.4483463160699 | 0.2390 |
| 66089 | 14.4432034979970 42.3074266229421 | 0.2029 |
| 73424 | 13.2636453804864 42.4675006580673 | 0.1996 |
| 35990 | 13.7905616382648 41.8109830098870 | 0.1981 |
| 16044 | 13.7436114375597 42.6123550872563 | 0.1976 |
| 87942 | 13.3391307998231 42.3547098998747 | 0.1802 |
| 6611 | 13.8817386009473 42.1996467035247 | 0.1690 |
| 45980 | 13.9329036425132 41.7757909721736 | 0.1558 |
| 87079 | 13.7914246030217 41.8098932751395 | 0.1485 |
| 87083 | 13.7914071282398 41.8093113608093 | 0.1484 |
| 73283 | 13.2633510504450 42.4675996999472 | 0.1456 |
| 87114 | 13.7910818999891 41.8081822999891 | 0.1445 |
| 87075 | 13.7912285170103 41.8100614001291 | 0.1420 |
| 45981 | 13.9325901556839 41.7758655392011 | 0.1412 |
| 87081 | 13.7913628516070 41.8095598485260 | 0.1305 |
| 87927 | 13.3396737815245 42.3547543222198 | 0.1276 |
| 24450 | 13.9353473815291 41.7752156564722 | 0.1271 |
| 33206 | 13.4647030869786 42.3258930064366 | 0.1257 |
| 74943 | 13.3826937064352 42.3494226012390 | 0.1236 |
| 16013 | 13.7436534001395 42.6124724000867 | 0.1220 |
| 48394 | 13.4679154591254 42.6319865247060 | 0.1197 |
| 16048 | 13.7422345447418 42.6122319611365 | 0.1154 |
| 52737 | 14.1659917501037 42.4259393999822 | 0.1141 |
| 70014 | 13.2701022825392 42.4484418526901 | 0.1140 |
| 46250 | 13.7913393295416 41.8109997679243 | 0.1080 |
| 73375 | 13.2628470000176 42.4700705000001 | 0.1079 |
| 45612 | 13.7983183441987 42.1101598916377 | 0.1043 |
| 45982 | 13.9327483418478 41.7758401092574 | 0.1031 |
| 45979 | 13.9330667114308 41.7757570874759 | 0.0964 |
| 74941 | 13.3801149335507 42.3515454579104 | 0.0948 |
| 73430 | 13.2625608000355 42.4652139000185 | 0.0937 |
| 26999 | 13.3520049413571 42.3656051819705 | 0.0925 |
| 87084 | 13.7913058647703 41.8093693195435 | 0.0919 |

### 3.3.2. Analysis of the Ranking of the Top-34 IBs

Table 6 summarizes the quantitative assessment of the goodness of the ranking of the top-34 IBs based on the value of metric *S*, against the manual classification. As it emerges, the algorithm classified optimally the top-34 IBs: *Recall = Precision = Accuracy =* 1.

The manual placement in the High-hazard class of the top-7 IBs of the Table 4 was easy for the following reasons. The buildings with ID 70015, 66089, and 73424 are very close to the riverbed (i.e., the value of *d* in Equation (2) is of few meters), so they are at high hazard of being flooded. About the remaining four buildings (ID: 35986, 35990, 16044, and 87942), they are located between 30 and 60 m from the river bed, but they are below the river's elevation (configuration 3 of Figure 1; $\Delta h_{b_i} = -10$ m) and,

therefore, they too are in a situation of high exposure to the hazard of being inundated. The remaining 27 IBs were placed in the Medium-hazard class because even though they are all at the same elevation as the river bed closest to them, they are at a distance from the river ranging between 50 and 90 m, and therefore they may be considered less exposed than the first 7 to flooding.

A further comment about the IBs located near the riverbed is the following. As mentioned, they are the ones most exposed to the hazard of being flooded, unless it happens that the building is separated from the river bed by a vertical wall (Figure 10 proposes an example). This circumstance that can only occur in areas with steep slope variations. This circumstance did not occur for the top-34 IBs of the case study.

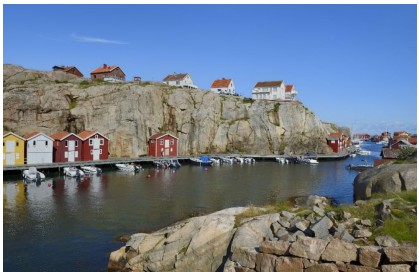

**Figure 10.** Buildings on the edge of a cliff.

**Table 6.** *Recall, Precision*, and *Accuracy* for the top-34 IBs of the Case Study.

|  | **Low** | **Medium** | **High** |
|---|---|---|---|
| *Recall* | Not Applicable | 1 | 1 |
| *Precision* | Not Applicable | 1 | 1 |
| *Accuracy* | Not Applicable | 1 | 1 |

## 4. Conclusions

The research summarized in this paper took place in five steps: (a) Formalization of a method for the computation of the ranking of the IBs located close to rivers in a given reference territory (that we called *GeoArea*); (b) creation of tables of a spatial DataBase and loading with the input data of the problem; (c) implementation of the proposed metric as SQL spatial queries; (d) experimentation of the method on a case study concerning buildings inside the Abruzzo region (Centre of Italy); (e) assessment of the computed ranking and, hence, a preliminary validation of the proposed method.

The two-steps method is applicable, with trivial modifications, to the ranking of IBs located near the sea. In such a case, the strip of respect has only one side: the sand side. This method is valuable for islands where the available land is a critical resource and the demand for tourism is growing, from that originates the drive towards the construction of abusive reception facilities close to the sea.

The final goal of our research is to implement a software that, fed with datasets about the *GeoArea*, the contour lines of the territory delimited by the *GeoArea*, the rivers crossing the *GeoArea*, and the buildings in the *GeoArea*, outputs the IBs (if any) as a ranked list. State government and local stakeholders fighting against the infringements of the urbanization planning rules are looking for an IT tool like this because it implements a numerical criteria on which a demolition strategy might be based. In countries like Italy the demolition of IBs is the only way to discourage future infringements of the law.

### 4.1. Cautionary Notes

Unauthorized buildings are a rapid phenomenon. There are evidence about dwellings built in a few days. It is, therefore, unrealistic to assume that there are up to date data available from institutional sites. A precondition to be satisfied because the tool we plan to develop becomes really

effective is that the dataset about the buildings inside the *GeoArea* is updated frequently (let say every three months). A way of obtaining updated data about the land use consists of the following steps (described in great detail in [25], Section 3): (a) acquisition of satellite raster data; (b) their processing through some of the known methods for automatic building detection (of high-resolution remote sensing images); (c) transformation of the raster data into vector data by means of a GIS software. Use this dataset to update the `Buildings` table of the SDB.

The ranking returned by the proposed two-steps method is affected by the *completeness* and the *quality* of the data about the IBs. Completeness and quality of geographic data are critical issues reported in almost all studies of the sector, e.g., [48]. In turn, the quality of the data about the IBs depends from two factors: the *accuracy* of the identification procedure of the suspicious buildings by using remote sensing methods and the *accuracy* of the raster-vector transformation step.

Our method is simple, the simplicity is paid in terms of reduced robustness. We conclude the paper by pointing out the method's weaknesses. The value of metric $S$ depends exclusively on parameters $P$, $d$, and $\Delta h$, since Equation (2) does not take into account the elevation profile of the terrain between the river and the building. This is an acceptable simplification of the geographical reality because, in our study, the buildings that are of interest are those located inside the strip of respect of rivers. In the case of the Italian Law no. 42, for example, the width of this belt of land is just 150 m. In a so narrow strip of land, the fluctuations of the ground elevation are necessarily moderate. Therefore, Equation (2) is not able to distinguish the geometric configurations for which it happens that the values of parameters $P$, $d$, and $\Delta h$ are identical. It follows that the buildings that are in these situations occupy the same ranking. Figure 11 proposes four different elevation profiles of the ground between a river ($r$) and a building ($b_1$). The value of $S$ and, hence, the position of $b_1$ in the ranking computed by Equation (2) is the same in the four cases, while the degree of exposure of the building to the flooding hazard is not the same in the four different situations. A way of refining the proposed method consists of taking into account the area under the "line" that connects the river to the building. The greater this area is, the greater the value of $S$ must be, because less will be the amount of water needed to reach the building. In the case of Figure 12, for example, the value of $S$ returned by Equation (2) is the same since, by hypothesis, the values of $P$, $d$, and $\Delta h$ are identical—while considering the area under the curve, it turns out that $b_1$ precedes $b_2$.

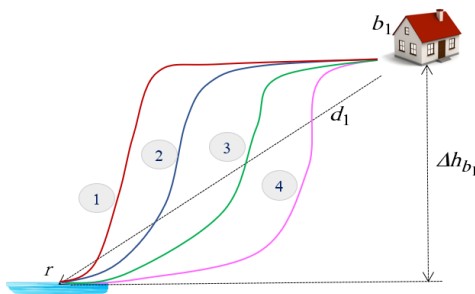

**Figure 11.** Four different elevation profiles of the ground between a river and a building.

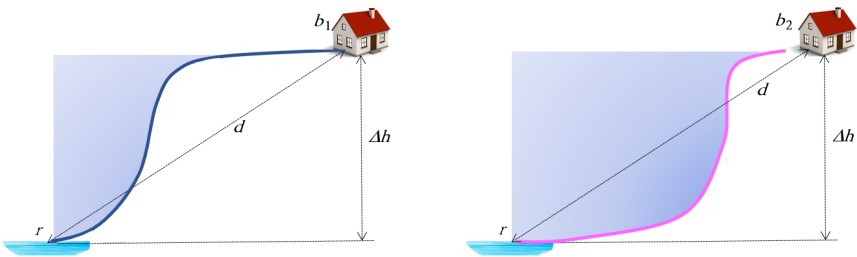

**Figure 12.** The profiles "2" and "4" of Figure 11.

From the above considerations, it arises the need to spend more time to validate metric *S*. The next step of our research will consist in carrying out a campaign of case studies applying, each time, the calculation of metric *S* to a different Italian municipality in order to collect statistics about *recall* and *precision*. Only after the *field* validation phase will it be clear whether it is necessary to make some adjustment to the algorithm for calculating *S* in order to increase its robustness.

**Supplementary Materials:** The following are available at http://www.mdpi.com/2220-9964/8/11/510/s1.

**Funding:** This research was funded by a grant from the University of L'Aquila.

**Conflicts of Interest:** The author declares no conflict of interest.

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
