# Peer review of "Ranking of Illegal Buildings Close to Rivers: A Proposal, Its Implementation and Preliminary Validation"

_ijgi, doi:10.3390/ijgi8110510_

Round 1

Reviewer 1 Report

Based on my understanding of the calculation of criterion S, it seems that the algorithm is not fully general, that the S criterion is too straight forward and does not take into account all possible situations to determine In the two cases illustrated in the attached document (case 1 and case 2), it is likely that the parameters required to calculate S are the same while the degrees of urgency involved in each situation are different. Can the authors indicate how these two cases are handled in their system? The validation procedure does not really validate criterion S. To complete it, flood simulations should be carried out in different configurations to see if the S values correctly sort buildings. The relevance of the content of sections 2.4 and 2.5 is questionable. The interest of providing SQL queries too. It is easy to understand that it is possible to integrate the necessary parameters to find the buildings. Figures 2, 3, 5 and 11 do not bring anything concrete to the understanding.

Author Response

C.#1

Based on my understanding of the calculation of criterion S, it seems that the algorithm is not fully general. The S criterion is too straightforward and does not take into account all possible situations to determine. In the two cases illustrated below (case 1 (left!!!) and case 2 (right!!!)), it is likely that the parameters required to calculate S are the same while the degrees of urgency involved in each situation are different.

Can the authors indicate how these two cases are handled in their system?

Reply

Case 2

Case 2 is composed of two scenes: Case 2-UP is the same as scene 1 of Fig.1; while the other one is the same as scene 4 of Fig.1. In both the geometric configurations, the building is sheltered from flooding episodes by a vertical wall of the same height that constitutes the natural barrier between the building and the river. For identical values of DeltaH (>0 in both scenes) and minimal Euclidian distance river-building, there is no reason why the two geometries of Case 2 have to be distinguished; so it is okay that the value of metric S is the same and, consequently, that the two buildings occupy the same position in the ranking.

Case 1

It is true that Eq.2 attributes the same value of S to the two scenes of Case 1, despite the degree of urgency is not the same. However, in the framework of the paper, this geometric configuration is not relevant as explained below.

The buildings that are of interest in our study are those located inside the Strip of Respect (SofR) of rivers. In the case of the Italian Law no.42, for example, the width of this belt of land is just 150 meters. But, for the peak (of Case 1-UP) to be high (as in the drawing of the reviewer) it is necessary that the house-river distance is extended, which implies that the buildings involved in these geometric configurations fall outside the SofR of the rivers, and therefore they are not illegal.

An alternative reply to the reviewer’s comment is the following.

Case 1 is the more critical the more DeltaH and the distance house-river take on low values. But for low values of the house-river distance (i.e. values of the order of a few tens of meters) the peak of the Case 1-UP is modest. Consequently, in the reality the two geometries of Case 1 look similar and, hence, it is not an issue whether the two buildings occupy the same position in the ranking.

C.#2

The validation procedure does not really validate criterion S. To complete it, flood simulations should be carried out in different configurations to see if the S values correctly sort buildings.

Reply

A good point, undoubtedly. However, the request of the reviewer is out of scope of our study. It, in fact, does not aim to relate urban planning and hydraulic risk management. A relevant topic discussed in many studies already published. Two examples: (Morelli et al., 2012) and (Luino et al., 2009).

Morelli. S. et al.: Urban planning, flood risk and public policy: The case of the Arno River, Firenze, Italy. Applied Geography 34 (2012) 205—218

Luino, F et al.: Application of a model to the evaluation of flood damage. Geoinformatica 13 (2009), DOI 10.1007/s10707-008-0070-3

C.#3

The relevance of the content of sections 2.4 and 2.5 is questionable. The interest of providing SQL queries too. It is easy to understand that it is possible to integrate the necessary parameters to find the buildings.

Reply

We understand the reviewer’s comment, but since the other three reviewers didn’t complain about these two small sub-sections we kept a merged version of them. The new sub-section (as the original ones) acts as the entry point to the Annex.

C.#4

Figures 2, 3, 5 and 11 do not bring anything concrete to the understanding.
Reply

Figures 2 and 5 (in the original manuscript) have been removed.

Reviewer 2 Report

The subject of the paper is original and the quality of presentation is high. The methodology and conclusions are clear. The paper can be published in its present form.

Author Response

He was satisfied.

Reviewer 3 Report

The issue of illegal buildings is very important in many countries. Although there have been many researches that have considered this topic, it is still a major issue for many Governments in the World how to manage legalization or demolition process. This includes in many cases remote sensing methods to identify buildings, but as it is suggested in the paper, there is no ranking to prioritize demolition process.  Therefore, the topic of the paper is relevant. However, the title and introduction are somewhat misleading since the paper is focused only on the buildings that are in the areas near the rivers or other water bodies, which are considered as protected zones and therefore construction work is prohibited. However, this is only a small portion of illegal buildings. For example, in my country of origin there are (according to official Government records) more than 2 million illegal buildings, and only a small percentage is near the rivers and other water bodies. It is not clear whether in Italy this type of illegal construction is most prevalent and whether exposure to the flood hazard is a major criterion for buildings to be demolished.

The mentioned law specifies the criteria by which the priority of demolition is ranked, but it is not clear from the developed method that these factors are taken into account. Where is the MCDM technique applied?

Is it possible to make Figure 12 clearer, since the mentioned water body is not visible? For example, draw a polygon around it, or mark it in some other way.

Table 5 is not illustrative enough. It gives geographic coordinates of the top ranked buildings. Is it possible to visualize these locations on the map in order to make the results clearer and unambiguous?

Author Response

C.#1

The title and introduction are somewhat misleading since the paper is focused only on the buildings that are in the areas near the rivers or other water bodies, which are considered as protected zones and therefore construction work is prohibited. However, this is only a small portion of IBs. For example, in my country of origin there are (according to official Government records) more than 2 million illegal buildings, and only a small percentage is near the rivers and other water bodies. It is not clear whether in Italy this type of illegal construction is most prevalent and whether exposure to the flood hazard is a major criterion for buildings to be demolished.

Reply

To better reflect the content of the research, I modified a bit the title of the paper. Moreover, the first sentence in the Introduction makes explicit the category of IBs the paper refers to.

C.#2

The mentioned law specifies the criteria by which the priority of demolition is ranked, but it is not clear from the developed method that these factors are taken into account. Where is the MCDM technique applied?

Reply

The application of the MCDM technique is outside the scope of the present paper, where the MCDM technique is mentioned as an effective way to sort the IBs according to Law no.S 580-B. In this frame, metric S (the focus of our proposal) can play the role of an environmental criteria.

C.#3

Is it possible to make Figure 12 clearer, since the mentioned water body is not visible? For example, draw a polygon around it, or mark it in some other way.

Reply

See Fig.9 (right).

C.#4

Table 5 is not illustrative enough. It gives geographic coordinates of the top ranked buildings. Is it possible to visualize these locations on the map in order to make the results clearer and unambiguous?

Reply

The (QGIS) map of Fig.8 (right) shows the location of the top-7 IBs (left) inside the Abruzzo region, as well as the corresponding S value.

Reviewer 4 Report

Author proposes the method of disclosure of informal settlement (illegal buildings, IBs) in the buffer along the watersheds and coastlines in the paper using the set of standard tools implemented into GIS closely coupled with DBMS (PostgreSQL) using the set of accessible (geographical) spatial data including newly acquired remote sensing images concerning the changing situation of built-up areas, which have to be vectorized before analysis.

The empirical evidence concerns Abruzzo  region. The value added to his methodology is proposal of introduction of Machine Learning to automate the verification  of the whole process of classification of the binary results of the ranking of IBs. This proposal is based on rigid law enforcement. 

Assuming the strict obey to law concerning the permission of building locations, which should be true in developed countries, it is no so clear in underdeveloped countries, where there can be a lot of additional factors to be taken into account, e.g. exceptions concerning houses on stilts or houses on the water (barges, even in the Netherlands) due to cultural traditions. 

However the comprehensive discussion of relevance of the problem is also the real added value of the paper.

There is only one weak step, an (undisclosed) assumption, that the vectorization and classification of images (transformation of raster images and implementation into geospatial data base) is made without any errors, which evaluation is not wider described in paper and treated as the standard procedure of "automatic building detection" outside of the described ranking of IBs procedure.

Author Response

C.#1

Assuming the strict obey to law concerning the permission of building locations, which should be true in developed countries,

<<The two reports by LegAmbiente mentioned in the paper state the opposite>>

it is no so clear in underdeveloped countries, where there can be a lot of additional factors to be taken into account, e.g. exceptions concerning houses on stilts or houses on the water (barges, even in the Netherlands) due to cultural traditions. 

<<Out of the study’s scope>>

C.#2

There is only one weak step, an (undisclosed) assumption, that the vectorization and classification of images (transformation of raster images and implementation into geospatial data base) is made without any errors, which evaluation is not wider described in paper and treated as the standard procedure of "automatic building detection" outside of the described ranking of IBs procedure.

Reply

The (new) sub section 4.1 touches the issue you mention above.

Round 2

Reviewer 1 Report

The ranking is expected to reflect the degree of exposure to the flooding hazard.

« Case 2 is composed of two scenes: Case 2-UP is the same as scene 1 of Fig.1; while the other one is the same as scene 4 of Fig.1. In both the geometric configurations, the building is sheltered from flooding episodes by a vertical wall of the same height that constitutes the natural barrier between the building and the river. For identical values of DeltaH (>0 in both scenes) and minimal Euclidian distance river-building, there is no reason why the two geometries of Case 2 have to be distinguished; so it is okay that the value of metric S is the same and, consequently, that the two buildings occupy the same position in the ranking. »

In case 2, the value of S is the same for both situations, while the amount of water needed to reach the house is much greater in the case of the bottom than in the case of the top (considering, of course, that the river is defined in the same way in both cases). The exposure is not equivalent for both situations.

« Case 1 is the more critical the more DeltaH and the distance house-river take on low values. But for low values of the house-river distance (i.e. values of the order of a few tens of meters) the peak of the Case 1-UP is modest. Consequently, in the reality the two geometries of Case 1 look similar and, hence, it is not an issue whether the two buildings occupy the same position in the ranking. »

If the goal is to define an order based on risk exposure, then it seems important to dissociate the variants.

« A good point, undoubtedly. However, the request of the reviewer is out of scope of our study. It, in fact, does not aim to relate urban planning and hydraulic risk management »

The simulation will determine whether the order determined by the algorithm based on S corresponds to the order in which the houses would be reached by a potential flood.

I can admit that the situations can be considered as approximately equivalent, but it seems to me, if they are not dealt with, that they are at least discussed and justified.

Author Response

Thanks for your further hints.

Sec.4.1 has been expanded.

Two more figures have been added.

Reviewer 3 Report

The author provided required answers.

Author Response

The English has been improved.

In the color paper the corrections are evident.

Thanks.